Closed-domain event extraction for hard news event monitoring: a systematic study

Dukić David david.dukic@fer.hr
Došilović Filip Karlo
Pluščec Domagoj
Šnajder Jan
University of Zagreb Faculty of Electrical Engineering and Computing , TakeLab , Zagreb , Croatia
Maguitman Ana
Electronic publication date: 2024 Oct 10
Publication date: 2024
Volume: 10
Electronic Location ID: e2355
Received 2024 May 17; Accepted 2024 Sep 2
Copyright: ©2024 Dukić et al.
Copyright year: 2024
Copyright holder: Dukić et al.
License: This is an open access article distributed under the terms of the Creative Commons Attribution License, which permits unrestricted use, distribution, reproduction and adaptation in any medium and for any purpose provided that it is properly attributed. For attribution, the original author(s), title, publication source (PeerJ Computer Science) and either DOI or URL of the article must be cited.
License URL: https://creativecommons.org/licenses/by/4.0/

Keywords: Event extraction, Hard news, Natural language processing, News event monitoring

Funding: AIDWAS KK.01.2.1.02.0285 This work was supported by the grant AIDWAS KK.01.2.1.02.0285. There was no additional external funding received for this study. The funders had no role in study design, data collection and analysis, decision to publish, or preparation of the manuscript.

==============================
News event monitoring systems allow real-time monitoring of a large number of events reported in the news, including the urgent and critical events comprising the so-called hard news. These systems heavily rely on natural language processing (NLP) to perform automatic event extraction at scale. While state-of-the-art event extraction models are readily available, integrating them into a news event monitoring system is not as straightforward as it seems due to practical issues related to model selection, robustness, and scale. To address this gap, we present a study on the practical use of event extraction models for news event monitoring. Our study focuses on the key task of closed-domain main event extraction (CDMEE), which aims to determine the type of the story’s main event and extract its arguments from the text. We evaluate a range of state-of-the-art NLP models for this task, including those based on pre-trained language models. Aiming at a more realistic evaluation than done in the literature, we introduce a new dataset manually labeled with event types and their arguments. Additionally, we assess the scalability of CDMEE models and analyze the trade-off between accuracy and inference speed. Our results give insights into the performance of state-of-the-art NLP models on the CDMEE task and provide recommendations for developing effective, robust, and scalable news event monitoring systems.

Introduction

Most information today exists in digital form, with over 90 percent of all available information being generated over the last couple of years (Tang, Ma & Luo, 2020). A substantial share of this information comes from online news stories (Foster, 2012), increasing our reliance on the Internet as a news source (Pew Research Center, 2019). At their core, news stories are arrangements and collections of event descriptions (Caswell & Dörr, 2018), typically conveying information about what happened, when and where, and who was involved. Some events, however, have a much larger impact on individuals and societies than others. For example, knowing about a critical event such as a flood in the neighboring area is arguably more important than finding out that a famous couple divorced today. Following this logic, journalists classify the news into hard news and soft news (Reinemann et al., 2012; Shoemaker & Cohen, 2012). Hard news includes urgent and critical events such as armed conflict, disease outbreaks, and natural disasters, which should be reported immediately. In contrast, soft news conveys information about hobbies, personal interests, lifestyle, and leisure topics, which typically need not be published at a certain time. We acknowledge that the definition of both hard and soft news is relative and can be subject to interpretation. What constitutes hard news for one individual or community may not hold the same significance for another. This subjectivity is influenced by cultural, social, and personal factors, as well as by the varying impacts of events on different groups. However, in this work, we align with the definition from Deng (2023) that hard news is something that needs to be reported immediately as it is of relevance to the broader public.

Although hard news has the potential to affect our lives radically, the large number of online news articles produced on a daily basis has long surpassed our sense-making capacity. News event monitoring systems aim to address this issue by offering real-time monitoring of events from online news, including news tracking (Osborne et al., 2014; Best et al., 2005), crisis event reporting (Tanev, Piskorski & Atkinson, 2008; Piskorski, Haneczok & Jacquet, 2020), and detecting and filtering fake news with the help of events (Zhang et al., 2023). Most such systems heavily rely on natural language processing (NLP) to perform automated event extraction (EE) from news articles at scale. Formally, EE is the task of detecting instances of events in the text and, for each event mention, identifying the event type and all the relevant aspects of the event (Liao & Grishman, 2010; Xiang & Wang, 2019). The EE task can be approached in two ways: closed-domain event extraction (CDEE), where there is a predefined set of event types, and open-domain event extraction, where the event types are not predetermined. News event monitoring systems typically rely on CDEE, focusing on a predefined set of urgent and critical events. Another useful simplification is the extraction of only the main event of each news article, which connects all other events in the story (Choubey, Raju & Huang, 2018). The CDEE task further consists of two subtasks: event detection (ED) and argument extraction (AE), typically arranged as a two-stage pipeline. The goal of ED is to locate and classify event mentions in the text, while the goal of AE is to extract the details of each event conditioned on the detected event type.

Recent advances in NLP, particularly in deep learning (DL) approaches to NLP, have made the EE task more feasible. Current state-of-the-art EE models utilize pre-trained language models (PLMs) based on the transformer architecture (Vaswani et al., 2017), leveraging the general linguistic competence of such models and adapting (“fine-tuning”) them to the EE task. The wide adoption of transformers and the availability of high-quality datasets gave rise to an array of EE models (Hamborg, Breitinger & Gipp, 2019; Yang et al., 2021; Zheng et al., 2019) and open-source toolkits (Yao, Peng & Zimu, 2022; Zhu et al., 2022; Hamborg, Breitinger & Gipp, 2019; Ma et al., 2021a).

With powerful EE models now being available off-the-shelf, it may appear that the state-of-the-art performance is up for grabs and that integrating EE into a news event monitoring system is rather straightforward. This, however, is not the case for at least three reasons. Firstly, due to subtleties in EE and the wide choice of EE models, it is far from obvious what models to select for the specific subtasks and how to combine them best. Secondly, off-the-shelf models are not robust enough to be applied directly to real-world data. A contemporary event monitoring system has to work on noisy data comprising diverse types of articles, not all of which pertain to events, let alone hard news. However, off-the-shelf models are trained on sanitized datasets and evaluated in idealized setups and are thus bound to perform suboptimal on real-world data. Lastly, scaling up EE based on DL models requires fast data processing with substantial computing power, usually provided by graphical processing units (GPUs). Typically, more accurate EE models will require more parameters and hence more compute time, thus reducing the inference speed and, consequently, the document throughput rate. On the other hand, smaller models, including the distilled variants of larger DL models, can run faster but typically achieve lower accuracies. This creates a trade-off between accuracy and inference speed, which should be taken into consideration when deciding which EE model is most suitable for production.

All this suggests that successfully integrating off-the-shelf state-of-the-art EE tools into a real-world news event monitoring system must address the practical issues of model selection, robustness, and scale. These issues, however, have so far been neglected in the literature.

Our work aims to fill the above-mentioned gap by presenting a study on the practical use of NLP event extraction models for event monitoring from hard news. Following Tong et al. (2022), we focus on closed-domain main event extraction (henceforth: CDMEE), which is arguably the most sensible setup for event monitoring from hard news. Our study comprises a systematic evaluation of state-of-the-art models for CDMEE from hard news articles, with the aim of answering three main research questions pertaining to model selection, robustness, and scale of CDMEE models for real-world news event monitoring:

RQ1: What is the performance of state-of-the-art NLP models for the CDMEE task (including ED and AE subtasks) from hard news articles?

RQ2: What is the performance of state-of-the-art CDMEE models in a realistic setup with noisy (i.e., non-hard news) data?

RQ3: What is the trade-off between CDMEE accuracy and inference speed across different CDMEE models and model variants?

To this end, we first evaluated and compared the performance of one traditional machine learning (ML) and a range of deep CDMEE models trained for ED and AE subtasks on hard news articles from DocEE (Tong et al., 2022), a standard CDMEE dataset for the English language. Our results reveal that on the ED subtask, the traditional ML models perform comparably to DL models, while on the AE subtask, question-answering DL models outperform sequence labeling models. To test the robustness of CDMEE models, a more realistic evaluation setup was introduced. This setup resembles the noisy input present in real-world news event monitoring systems. For that purpose, we compiled ASHNEE, a novel dataset with document-level event type and argument labels. Unlike other publicly available EE datasets, ASHNEE also includes non-hard news articles, making a realistic evaluation of CDMEE models’ performance on noisy input possible. The results show that, while optimal performance can be achieved by considering the hard, soft, and event-less articles as separate classes when training CDMEE models, in a noisy setup, the performance on the AE subtask typically deteriorates more than on the ED subtask. Lastly, we analyzed the accuracy-speed trade-off of ED and AE models run on a GPU, as well as speed reductions in their CPU counterparts. We show that distilled CDMEE models can perform on par with their full-model counterparts for the ED subtask, while medium-sized models, when run on a GPU and with larger batch sizes, are suitable for both EE tasks.

Our contributions can be summarized as follows: (1) A novel CDMEE dataset consisting of hard news articles with manually labeled event types and arguments as well as non-hard news articles, allowing for a more realistic evaluation of news event monitoring systems; (2) Benchmarking of a range of state-of-the-art NLP methods for the CDMEE task, and (3) Investigation of the robustness and scale issues by performing a realistic evaluation of CDMEE models and analyzing the accuracy-speed trade-off. A high-level overview of the proposed hard news event monitoring system architecture, based on the findings from our study, is shown in Fig. 1. We believe our findings will guide the event extraction community and industry toward building more effective, robust, and scalable event monitoring systems from hard news.

Figure 1 Proposed hard news event monitoring system architecture.

Figure was created using draw.io and all the templates, graphics, and silhouettes were created using icons built into the draw.io web application.

The rest of this article is structured as follows. The article starts with a detailed overview of related work in Section ‘Related work’. Our study is presented in Sections ‘Methodology’ and ‘Experiments’. Specifically, the ASNEE dataset, the CDMEE models, and evaluation setups are described in Section ‘Methodology’ and the experimental results are presented in Section ‘Experiments’. Finally, we discuss the results in Section ‘Discussion’ and conclude with suggestions for future work in Section ‘Conclusion’.

Background and Related Work

The wide availability of EE datasets and the maturing of DL technology triggered a proliferation of diverse toolkits and a range of EE applications. This section gives an overview of the state-of-the-art in closed-domain EE, open-source toolkits for EE, and large-scale information extraction systems that rely on extracted event attributes.

Approaches to event extraction

As mentioned in the introduction, EE can be performed in either a closed-domain (CDEE) or open-domain fashion. The former aims to identify events and their attributes, relying on a predefined set of events (Li et al., 2024), while the latter infers the event schema and unconstrained event types from the data directly (Xiang & Wang, 2019). Although open-domain event extraction is generally useful for discovering new event types (Liu, Min & Huang, 2021; Liu, Huang & Zhang, 2019), most EE applications are domain-specific, and the gain from the application perspective is the highest when EE operates on a closed set of events (Piskorski, Haneczok & Jacquet, 2020; Best et al., 2005). Specifically, narrowly restricted domains, such as the news domain (Piskorski, Haneczok & Jacquet, 2020) or the biomedical domain (Ramponi et al., 2020), may benefit from a more fine-grained structure of a smaller closed set of events.

Another distinction, based on the level at which an EE model operates, is between sentence-level EE and document-level EE. In the former, the main assumption is that events and arguments coexist in a single sentence. In contrast, document-level EE makes a more realistic assumption that events and their corresponding arguments may be scattered across multiple sentences, where event mentions are potentially linked by event coreference relations. Recent research suggests that sentence-level EE is more suitable for highly specific domains, such as the biomedical and cybersecurity domains (Ramponi et al., 2020; Man Duc Trong et al., 2020). In contrast, document-level EE appears to be a more sensible choice for news event monitoring (Hamborg, Breitinger & Gipp, 2019; Tong et al., 2022). This is corroborated by the recent availability of news-based datasets for document-level EE (Tong et al., 2022; Ebner et al., 2020; Han et al., 2022). Moreover, there has been a recent shift toward a more application-oriented variant of document-level EE, namely one that aims to extract only the main event of a news article (Hamborg, Breitinger & Gipp, 2019; Tong et al., 2022), where the main event is defined as the most dominant event in a news article that governs and connects to other mentioned foreground and background events (Choubey, Raju & Huang, 2018). Our study focuses on this approach to EE, which we refer to as closed-domain main event extraction. Arguably, CDMEE is the most sensible EE approach for event monitoring from hard news, considering that urgent and critical events typically constitute the main events of news articles.

Typically, CDEE (also CDMEE) is performed in a pipelined setup, where the results of the ED module are fed to the AE module (Ahn, 2006). Each news article is assigned an event type inherited from the event type of the main event described in the article, where the types come from a predefined set. The extracted arguments are also predefined and depend on the event type. Typical argument roles include, but are not limited to, event participants, locations, and instruments. The CDEE setup allows the users of an event monitoring system to sift through text data by searching for mentions of predefined event types and arguments.

Document-level EE models

Current state-of-the-art document-level EE models are Pseudo-Trigger-aware Pruned Complete Graph (PTPCG) (Zhu et al., 2022), Document-to-Events via Parallel Prediction Networks (DE-PPN) (Yang et al., 2021), and Document To Entity-based Directed Acyclic Graph (DOC2EDAG) (Zheng et al., 2019). These DL models achieve their best results on the ChFinAnn dataset (Zheng et al., 2019). The highest scoring model is PTPCG, which achieves an F1 score of 79.4 and is closely followed by DE-PPN and Doc2EDAG, with F1 scores of 77.9 and 76.3, respectively. The DE-PPN uses an encoder–decoder architecture to extract events and their arguments in an end-to-end fashion. DOC2EDAG and PTPCG rely on entity recognition and reframe the document-level EE task as a multi-task problem, where Zheng et al. (2019) employ transformers and Zhu et al. (2022) use the traditional recurrent neural network architecture. While state-of-the-art, these models are complex, cumbersome to run and deploy in production, and cannot easily leverage PLMs.

EE toolkits

One of the EE toolkits suitable for news event monitoring is GiveMe5W1H (Hamborg, Breitinger & Gipp, 2019), which extracts answers to journalistic 5W1H questions (What happened?, Who was involved?, When did the event occur?, Where did the event occur?, Why did it happen?, and How did it happen?). The system relies on syntactic clues, specifically named entities and part-of-speech tags, and pattern matching to find candidate answers. The hand-made heuristic function assigns a score to a candidate answer, and the highest-scoring answer is selected as the answer to a particular question. The GiveMe5W1H is simple, interpretable, and can run on CPU hardware. However, the system has not been adequately evaluated, and the need to write a heuristic for each category hinders its maintainability and extensibility.

EventPlus (Ma et al., 2021b) is a pipeline for analyzing temporal relations between events extracted from a single document. The system was tested on the news (ACE 2005 dataset; Doddington et al., 2004) and biomedical domain (GENIA dataset; Ramponi et al., 2020). It uses several methods to extract events from multiple domains, search for arguments across the entire document, and an additional model to handle speculative events and negation. However, the system is intended for sentence-level event extraction and, as such, is not directly applicable to news event monitoring, where the system needs to identify and extract only the main event in each article.

DocEE1 (Zhu et al., 2022) is another toolkit for document-level EE. In contrast to document-level main event extraction, DocEE extracts all events from a document, treating EE as a sentence-level task. The toolkit supports different task-specific models of varying complexity and size, making it suitable for various applications.

The latest in a series of open-source EE toolkits is OmniEvent (Yao, Peng & Zimu, 2022). This toolkit supports both older DL models and modern, transformer-based EE models. For the AE subtask, the toolkit provides a number of standard approaches, namely sequence labeling, question answering, and sequence-to-sequence. Several models were already tested on different datasets, including ACE 2005, MAVEN, and DuEE datasets (Ahn, 2006; Wang et al., 2020; Han et al., 2022), allowing system designers to make an informed decision on which models to use for their use-case.

Most toolkits mentioned above work at the sentence level, use complex, task-dependent models, or do not use DL altogether, which complicates their use in news event monitoring systems that require robust EE models. Even the state-of-the-art document-level CDEE models would require an additional step of aggregating the extraction results and selecting the main event and its corresponding arguments. Our study aims to address this problem by employing a standard DL arsenal on document-level ED and AE models for CDMEE and examine the applicability of such an approach for news event monitoring.

News event monitoring systems

European Media Monitor (https://emm.newsbrief.eu/NewsBrief/clusteredition/en/latest.html) (EMM) (Atkinson & Van der Goot, 2009; Best et al., 2005), a large-scale online news and social media monitoring system developed and maintained by the European Commission’s Joint Research Centre, is perhaps the most elaborate example of an event monitoring system powered by EE methods. The system aggregates news articles from dozens of languages, reporting on events around the globe in real-time. The event extraction system is part of the EMM’s information extraction infrastructure, which relies on different linguistic and syntax cues, extracting event information from a cluster of articles (Erik, 2008).

A more recent event monitoring system is Liveuamap (Live Universal Awareness Map), (https://liveuamap.com/) which displays events reported around the globe, with a focus on conflicts. Even though the system automatically crawls and extracts information from news sources, Liveuamap employs editors to filter the relevant events before they are available on the map.

Other EE systems tend to be more specialized. For example, Timelines (Piskorski et al., 2020) emphasizes entities, entity-event relations, and events related to a specific entity. On the other hand, FrontEx (Piskorski & Atkinson, 2011) focuses on border-related events, e.g., illegal immigration and crisis situations.

Although the PLMs fine-tuned on small datasets can achieve satisfactory performance, the need for human-collected and aggregated event data is still present. One such effort is Wikipedia’s Current Events Portal (WCEP), (https://en.wikipedia.org/wiki/Portal:Current_events) a crowdsourced news summary portal where users can add and summarize events linking to the source of the event reported in a news article. WCEP was successfully used to create a large-scale dataset for multi-document event summarization (Ghalandari et al., 2020). Another specialized portal is the Armed Conflict Location & Event Data (ACLED) (Piskorski, Haneczok & Jacquet, 2020) (https://acleddata.com) which collects information about armed conflicts around the world.

The systems mentioned above were either built for particular purposes (e.g., ACLED) or applied to various sources inside a specific domain (e.g., EMM). Our study tries to attain a level of generality that encompasses all these particular domains. Thus, instead of focusing on specialized and domain-specific model architectures, we investigated whether current state-of-the-art EE models can be used for the CDMEE task and, in turn, for real-world news event monitoring applications. The success of such a general approach could facilitate the broader adoption of CDMEE in practice.

Methodology

In this section, we describe our study’s methodology by describing datasets, models, and model evaluation strategies for CDMEE.

Datasets

The task of a CDMEE model is to determine the main event described in the article and extract the associated arguments. The success of solving this task hinges on the quality of the data used to train the model. At the time being, the only document-level EE dataset suitable for data-hungry DL models is DocEE (Tong et al., 2022). We used the DocEE dataset to fine-tune our models, while for model evaluation, we also used the ASHNEE dataset, a new dataset we compiled specifically for that purpose.

DocEE dataset

The DocEE dataset of Tong et al. (2022) is the first document-level EE dataset intended and suitable for real-world news event monitoring applications. The dataset contains over 27K documents, sampled from Wikipedia and news articles (BBC, ABC News, and The Guardian, among others), comprised of 31 hard news and 28 soft news event types.2 The dataset covers a diverse set of event types, including Armed conflict, Earthquakes, Fire, Floods, New archeological discoveries, and Sports competition. The dataset contains 351 argument types, and more than 180K argument mentions. The number of unique argument types per event type ranges from 4 to 21, with an average of nine argument types. The size of the dataset—in terms of the number of events, articles, and fine-grained arguments—makes it an ideal dataset for developing DL-based CDMEE models for news event monitoring.

Although DocEE is quite suitable for CDMEE, we have identified two problems with using it for robust news event monitoring. First, DocEE contains articles that are too clean to be used as a basis for a realistic evaluation of EE systems. Namely, unlike the cleaned-up DocEE articles, online news articles are typically automatically scraped and may contain noisy artifacts such as in-content advertisements and links to other articles. It is unclear how DocEE can be used to train models that can handle the noise that occurs in articles in the wild. Secondly, and more importantly, the DocEE dataset does not contain event-less articles, i.e., news articles that do not describe an event per se. Online news outlets, however, publish plenty of such articles, including opinion pieces, sponsored articles, and advertisements. This means that a real-world EE system must be capable of distinguishing between eventful and event-less articles. DocEE, however, cannot be used to train such systems as it only contains articles that report on events. All this points to the need for a dataset collected from real-world news article data.

ASHNEE dataset

To make up for the shortcomings of the DocEE dataset, we created a benchmark dataset intended for a more realistic evaluation of CDMEE models, specifically CDMEE models for EE from hard news articles. We call our dataset ASHNEE (Automatically Scraped Hard News Event Extraction). We argue ASHNEE is perfectly suitable for evaluating models for detecting eventful articles in the wild and evaluating CDMEE models on noisy data.

ASHNEE contains articles scraped from 577 news outlets in English (most of which are from the USA, the UK, and Australia) published between 2019 and 2022. The dataset contains ground-truth labels for document-level closed-domain ED and AE on hard news articles with additional labels for soft news and event-less articles labeled as other. We inherited the DocEE event taxonomy, including the event arguments. For more details, see the official repository (Supplemental File S2).

Data labeling was carried out in two phases. In the first phase, we labeled the event types for the scraped articles, and in the second, we labeled arguments for scraped hard news documents. Six coders, near-native speakers of English, participated in these tasks. To ensure the quality of the labels, each article was labeled by two coders. The idea of labeling the main event came from DocEE. Before annotation, we calibrated our coders on the news portion of the DocEE dataset. The data was annotated in two rounds. First, we used a model trained on the DocEE dataset to strategically sample instances for labeling from a pool of automatically scraped news articles. The goal was to use the model to pre-label the articles and then select those that belong to hard news events. In this way, we minimize the labeling cost and the number of articles that belong to soft news or event-less articles. The coders were unaware of this step or the label the model predicted. For articles that describe multiple events, the coders were instructed to label the predominant event as the article’s main event. If the main event of the article did not match any event in the event taxonomy (e.g., in the case of soft news) or the article did not contain an event at all, the article was labeled with the class other. The articles that belonged to this class were soft news, but mostly advertisements and opinion pieces, i.e., event-less articles. Therefore, after the first round, we trained the model again, including the examples from the class other, and resampled the articles from the pool. In this way, we could select more eventful articles and reduce the likelihood of ending up with too many event-less articles. The inter-coder agreement (Cohen’s kappa coefficient) was 0.71, which is considered a substantial agreement (Landis & Koch, 1977). For the final version of the dataset, we retained only the articles for which both coders agreed on the event type.

In the second phase, we asked the coders to label the arguments for the hard news events obtained in the first phase. Since this phase was harder, we increased the pool of coders to eight. To reduce the cognitive burden on the coders, we divided the event types into four groups, each containing 80 arguments. This way, all coders assigned to the same group had to focus on a smaller set of arguments, i.e., only on the arguments of the event types that were present in the group. Labeling of arguments was carried out in two rounds. We assigned two coders to each group, and two coders labeled each document in the group. First, we calibrated the coders on a news portion of the DocEE dataset, as we did in the first phase. After calibration, for the first labeling round, we gave the coders labeled articles from the ED phase to label arguments in the articles. Finally, for the second labeling round, we assigned two coders with experience in both event-type labeling and event argument labeling to resolve any labeling disagreements.

In total, the coders labeled 2,279 articles, of which 1,050 as hard news articles from one of 26 hard news event types4 and 1,229 as event-less (the label other). The final dataset contains 6,058 argument mentions and 158 unique argument types. Figure 2 shows an example of an article in which the arguments of the main label Earthquake have been labeled in text.

Figure 2 An example article from the ASHNEE dataset, featuring an event of type Earthquakes, with argument labels shown in the text.

Models

Our study tested one traditional ML model for ED and a range of DL models for both ED and AE. The traditional ML model requires fewer computing resources and may thus be a more suitable choice for real-time news event monitoring, albeit at the cost of decreased accuracy. PLMs have already proven as DL models of choice for attaining state-of-the-art CDEE results (cf. Section Related work). We do not include recent models from the GPT family due to their poor zero-shot performance on event extraction (Gao et al., 2023). Even the largest models of the GPT family, such as ChatGPT, fall far behind state-of-the-art results in information extraction tasks (Han et al., 2023). This disparity becomes more pronounced when compared to smaller, encoder-based models. Recent work by Dukić & Šnajder (2024) has shown that decoder-only models from the GPT family lag behind drastically smaller encoder models in information extraction tasks, even when fine-tuned with instruction tuning (Mishra et al., 2022). Additionally, adapting GPT-like models to identify event types and arguments based on a predefined CDEE schema requires extensive fine-tuning, since these models are typically comprised of billions of parameters. These models are also resource-intensive, as they cannot operate efficiently on CPUs, making them unsuitable for real-time applications where rapid predictions are needed for a continuous stream of incoming documents.

Event detection

The traditional ML ED model we tested is a linear support vector machine (SVM) model with term frequency-inverse document frequency (TF-IDF) weighted tokens as features. We tokenized and lemmatized the text on both DocEE and ASHNEE datasets and used unigrams, bigrams, and trigrams of lemmatized tokens as TF-IDF features. Our implementation uses the linear SVM (with hyperparameter C = 1) and TF-IDF from the scikit-learn (https://scikit-learn.org/stable) library and spaCy (https://spacy.io) for tokenization and lemmatization. The model was trained on DocEE and evaluated on the test set of DocEE and ASHNEE datasets. We consider this model as the ED baseline in our study.

Relying on an available arsenal of models and approaches, we can employ existing transformer-based PLMs for the ED subtask. Transformer-based PLMs can easily be used for ED, modeling the problem as a multi-class text classification task, similar to the models from Tong et al. (2022) and Piskorski, Haneczok & Jacquet (2020). We tested four transformer-based PLMs for ED: two standard PLM models, namely RoBERTa (Liu et al., 2019b) and DeBERTa (He, Gao & Chen, 2021), and two distilled models—smaller transformer-based language models trained to mimic the behavior of larger models but with fewer parameters—namely, ALBERT (Lan et al., 2019) and DistilRoBERTa (Sanh et al., 2019).

For AE, we again used transformer-based PLMs, but tested two standard framings of the task (Li et al., 2024): sequence labeling and question answering. Model implementations from the Hugging Face (https://huggingface.co) library were used.

Sequence labeling-based argument extraction

The AE subtask can be framed as a token classification task where predefined event arguments are sequences of words in the text, similar to the sequence tagging methods of Tong et al. (2022) and Du & Cardie (2020a). The model is trained to predict, for each token in a document, whether the token is part of a predefined argument type or outside of it, using the IOB2 label scheme (Ratnaparkhi, 1998). However, as each event type has its arguments, the model has to capture the relation between the event type and its arguments. We experimented with two model variants: explicit and implicit. Both are conditioned on the event type, but the relation between arguments and event type is modeled either explicitly via multiple classification heads or implicitly via event type embeddings (the neural representations of event types learned from scratch). To the best of our knowledge, both approaches for modeling the relation between event types and their arguments are novel, as previous sequence labeling approaches to AE did not model the connection between event types and their arguments.

In the explicit case, each event type has its classification layer with the number of possible outputs corresponding to the number of arguments for that event type. We trained each token classification head to specialize in AE conditioned on the specific event type. This can be viewed as an instance of the more general multi-task learning setup (Zhang et al., 2022). In contrast, the idea behind the implicit case was to initialize an embedding matrix for each event type and use it as a trainable lookup table, similarly to Pouran Ben Veyseh & Nguyen (2022) and Liu et al. (2019a). The embedding matrix was trained together with all other model parameters, with each row of the matrix mapping to one event type embedding. The embedding matrix row corresponding to the input text main event was concatenated to the output of the RoBERTa (Base) model on each token position before passing it to the dropout and the fully connected classification layer. This way, the model was implicitly forced to learn that the tokens from the input document belong to one event type.

Unlike previous work on sequence labeling for AE (Boros, Moreno & Doucet, 2021; Du & Cardie, 2020b; Li, Ji & Han, 2021; Tong et al., 2022), which all used BERT as the base model, we opted for RoBERTa (Base), which has been shown to outperform its predecessor BERT on many NLP tasks (Liu et al., 2019b).

Question answering-based argument extraction

For the second variant of the AE model, following the works of Li et al. (2020), Liu et al. (2020), and Du & Cardie (2020b), we framed the problem as extractive question-answering (QA). The goal is to find the answer (a span of text) to a question, where each question corresponds to a particular argument. Each training example is a triple comprising a context, a question, and an answer. The context corresponds to the article’s text, the question corresponds to a specific event-type dependent argument, and the answer corresponds to the argument text span. The answer is an empty string if the text does not mention the corresponding argument. We used the models pre-trained on the SQUAD-v2 dataset (Rajpurkar, Jia & Liang, 2018), a standard dataset for evaluating extractive QA models, and fine-tuned them on DocEE arguments. Models return character-level positions of the answer. To use these models, we first tokenized the examples using the model’s pre-trained tokenizer. We then processed the tokenized examples to label the start and end positions of the answers in the text. If the context was too long to fit into a single example, we broke it down into multiple examples, each overlapping with the previous one. Finally, we returned tokenized examples with their labeled start and end positions.

The two model variants for the AE subtask—sequence labeling and question-answering—deserve additional comments. Each method has its advantages and disadvantages. The sequence labeling method is a straightforward way to model the AE subtask if one does not need to model the relationship between the event type and its arguments. However, modeling this relationship introduces additional complexity, as described above. Another problem is due to the overlapping arguments, the modeling of which would introduce even more complexity. Furthermore, because PLMs have a limit on the number of input tokens, sequence labeling methods built on top of these models cannot model arbitrarily long arguments. On the other hand, the extractive QA framework has shown to be a more effective alternative—integrating information about event types and dealing with long texts is rather straightforward with extractive QA. Here, however, multiple occurrences of the same argument could pose a problem. This can be addressed in several ways. One option is to aggregate the occurrences and treat them all as possible and equally valid argument mentions for a single argument type. Another option is to aggregate the occurrences and extract multiple argument mentions. Yet another option is to take only the first-occurring argument mention in the text, assuming the first mention will be the most descriptive one. We chose the first option for our experiments.

Datasets preparation

We split the DocEE dataset into a training and a test set in a 4:1 ratio for our experiments. We randomly sampled a validation set from the training data for early stopping and hyperparameter tuning. We used all articles from the DocEE dataset and all hard news articles plus other articles from the ASHNEE dataset. As each model required specific dataset preprocessing, we detail the steps employed for each model and dataset.

For the ED subtask, we concatenated the title and body fields of the DocEE articles into a single field to maintain consistency with the ASHNEE dataset, which does not contain a separate title field. Additionally, PLMs have a limit on the number of tokens, so the examples longer than the maximum number of tokens had to be truncated.

For the AE subtask, as the DocEE dataset has character-level annotations of argument positions, those needed to be converted to token positions before applying the sequence labeling model. We aligned character-level argument spans with token-level spans with the help of the spaCy tokenizer.5 As the transformer-based sequence labeling models have a limit on the number of input tokens, the examples longer than the limit (512 tokens) had to be truncated. While this limitation may negatively affect model performance, the effect is limited by the fact that 90% of all arguments in the DocEE dataset appear in the first 512 tokens. The input documents were not lowercased. The ASHNEE dataset was not used for sequence labeling AE experiments.

For QA-based AE, we had to preprocess the DocEE and ASHNEE datasets to match the structure of the SQUAD dataset. The datasets were prepared by extracting a triple comprising the context, question, and answers. As some argument types occur multiple times in a document, the corresponding question has multiple possible answers. This yielded 103, 130 and 5, 618 question-answer pairs for the DocEE and the ASHNEE datasets, respectively.

Evaluation

In line with standard practice for evaluating ED models (Li et al., 2024; Lai, 2022), our experiments use precision, recall, and F1 metrics. Specifically, because the datasets are imbalanced, but each class is considered equally important, the macro variants of these metrics are used.

For AE sequence labeling models, we followed the standard practice for evaluating sequence labeling models (Nakayama, 2018) and used the seqeval (https://github.com/chakki-works/seqeval) library with strict matching in conjunction with the IOB2 label scheme. We report micro precision, recall, and F1 score, as implemented in Hugging Face’s evaluate library (https://github.com/huggingface/evaluate/blob/main/metrics/seqeval/seqeval.py).

To evaluate the QA-based AE models, we used the standard metrics described in Rajpurkar et al. (2016); Rajpurkar, Jia & Liang (2018) and implemented in Hugging Face’s evaluate (https://huggingface.co/spaces/evaluate-metric/squad_v2) library. Three evaluation scores are reported: two evaluation scores that pertain to the model’s performance on positive examples (the arguments that are present in the text) and one evaluation score that takes into account the overall model’s performance, taking into account also the negative examples (the arguments not present in the text). The performance is evaluated by taking the average overlap of tokens between the predicted and the ground-truth arguments. As some arguments may have several mentions in one article, the corresponding question has multiple possible answers. In such cases, the predicted answer is evaluated against each possible ground-truth answer, and the best-scoring pair, measured in terms of micro F1, is reported in the overall evaluation.

Experiments

This study aimed to systematically evaluate state-of-the-art CDMEE models for event monitoring from hard news articles. We focused on the questions of model selection, robustness, and scale, which we consider of immediate practical relevance for real-world event monitoring systems. Specifically, to address our research questions (cf. Section Introduction), we (1) evaluated the performance of baseline and state-of-the-art CDMEE models for ED and event AE, comparing sequence labeling and question-answering variants; (2) tested how CDMEE models perform in the presence of noisy articles and how to adapt them to discriminate between hard and non-hard news articles; and (3) investigated the accuracy-speed trade-off of CDMEE models for both ED and AE.

As described in Section Introduction, a typical EE pipeline comprises two modules: ED and AE, where the predictions of the former are fed as inputs for the latter. Our experiments followed separate evaluations for these two modules, using both ground-truth and predicted ED labels for AE training and evaluation. We compared the traditional ML and deep learning model variants’ performance on DocEE and ASHNEE datasets.

Event detection

We start by reporting the performance of different classification models on the hard news portion of the DocEE dataset. We then explore how the models performed on new and noisy data from the ASHNEE dataset.

All models were trained and evaluated on the hard news portion of the DocEE dataset. We sampled 10% of the training data and used that as a validation set. Models were trained for a maximum of 10 epochs with batches of 64 examples. We set the early stopping threshold to 10 steps, evaluating the performance on the validation set every 20 updates.

Table 1 shows the ED performance of different ED models, namely SVM, RoBERTa, DeBERTa, ALBERT, and DistilRoBERTa. The best-performing model for document-level ED was RoBERTa (Large) with a macro F1 score of 94.70%. The performance gap between the best-performing model and linear SVM is 3.20%, and, from a practical viewpoint, suggests that one could resort to linear models for state-of-the-art performance. Results also show that distilled transformer-based models, namely ALBERT and DistilRoBERTa, can achieve competitive performance, suggesting they could be used instead of larger PLMs.

Table 1 Event detection results for hard news events from the DocEE dataset.

The best results are in bold.

Model	Precision (%)	Recall (%)	F1 (%)	
SVM (Linear)	92.90	90.80	91.50	
RoBERTa (Base)	95.50	93.80	94.50	
RoBERTa (Large)	95.40	94.10	94.70	
DeBERTa-v3 (Base)	93.20	91.00	91.70	
DeBERTa-v3 (Large)	94.40	93.50	93.80	
ALBERT-v2 (Base)	94.90	93.30	93.90	
DistilRoBERTa (Base)	94.80	93.80	94.20	

To make the results comparable between ASHNEE and DocEE datasets, we evaluated the models trained on the DocEE dataset only on the hard news portion of the ASHNEE dataset, ignoring the articles belonging to the class other. Table 2 shows the ED performance of text classification models trained on the DocEE dataset and evaluated on the ASHNEE dataset. Results show a 4.30% drop in performance for RoBERTa (Large), the best-performing model for the DocEE dataset. The distilled models experienced the worst performance drop of 6% F1 points. The performance drops were to be expected as the models were not trained with examples from the ASHNEE dataset. However, the performance decreases were not severe enough to indicate that the models lack generalization capabilities when presented with noisy data.

Table 2 Event detection results for hard news events on the ASHNEE dataset.

The best results are in bold.

Model	Precision (%)	Recall (%)	F1 (%)	
SVM (Linear)	90.60	91.70	90.20	
RoBERTa (Base)	87.90	93.00	89.60	
RoBERTa (Large)	89.90	91.80	90.40	
DeBERTa-v3 (Base)	92.30	95.40	93.20	
DeBERTa-v3 (Large)	82.80	89.30	83.70	
ALBERT-v2 (Base)	86.50	91.50	87.90	
DistilRoBERTa (Base)	87.30	90.60	88.00	

Detecting hard news articles

A real-world event monitoring system for hard news must distinguish between articles about hard news and all other articles, including soft news and event-less articles, such as opinion pieces, sponsored articles, and advertisements (as we propose in Fig. 1). The naive approach would be to train a binary classifier that predicts whether each document belongs to hard or soft news. We trained a binary classifier using the DocEE training set, building on top of RoBERTa (Base). It achieved a macro F1 score of 96.82% on the held-out DocEE test set, indicating that this PLM easily discriminates between hard and soft news articles. However, in a realistic scenario, detecting hard news involves distinguishing between hard news articles and other articles, where other encompasses not only soft news but also articles without events. In the context of closed-domain main ED, which is inherently a multiclass problem, distinguishing between hard news events and all other cases ultimately depends on how non-hard news articles are labeled. We considered two options:

• Hard-vs-other: The first option is to discriminate between all hard news types on one side and all other articles on the other, effectively merging event-less and soft news articles into a single category other. The approach requires fewer classes, which in principle should make the model easier to train, but the category other covers very heterogeneous articles, which makes the classification problem harder;

• Hard-vs-soft-vs-other: The alternative is to treat each soft news event type as a separate class and event-less articles as their own class. This approach requires more classes, which will presumably be more homogenous.

We tested both approaches using two multi-class text classification models built on top of the RoBERTa (Base), which we refer to as either hard-vs-other or hard-vs-soft-vs-other. For the hard-vs-other model, we combined all soft news articles from DocEE with the other articles from ASHNEE into a common other class while retaining the original labels for DocEE hard news articles. This approach was taken because ASHNEE contains a few soft news articles, due to the way it was compiled. In contrast, we trained the hard-vs-soft-vs-other model to differentiate between DocEE events (both hard and soft) and the ASHNEE other articles. To make the two models comparable, after obtaining the predictions of the hard-vs-soft-vs-other model, we assigned all soft news predictions the label other.

The hard-vs-other model achieved a macro F1 score of 90.54%, while hard-vs-soft-vs-other model achieved 91.96%. Although this is not a considerable difference, it indicates that it is better to differentiate between soft, hard, and other articles than to merge soft and other articles into one class. Furthermore, this way, we can have one model for both detecting and classifying the hard news articles if we can afford to sacrifice a bit of performance, as opposed to having a separate hard news detector and a separate event detection model for hard news event type classification.

Argument extraction

Here, we present the training details and results for AE models in both sequence labeling and QA-based variants.

Training details

Figure 3 shows a sentence sampled from a DocEE document labeled with arguments at the token level. Recall from Section Methodology that the model we use for sequence labeling is the RoBERTa (Base) transformer with a token classification head, where the event type information is incorporated either explicitly via multiple classification heads or implicitly via event type embeddings. The hyperparameter tuning was performed on the validation set, which consisted of 20% of the training data. The embedding matrix was randomly initialized, and we used a higher learning rate for the parameters making up this matrix than for the PLM backbone parameters. The intuition behind this is that we want the embedding parameters to update faster during training because, unlike PLM parameters, the randomly initialized embeddings do not encode any language knowledge. We experimented with several learning rates and tuned the event embedding matrix dimension. The implicit model was trained on the train split of the DocEE dataset. The learning rate for the embedding matrix parameters and the embedding dimension was optimized via grid search with a fixed seed across runs. The best model from the 10 epochs was picked out based on micro F1 performance on the validation set for each hyperparameter combination. The training setup for the explicit model was identical, except no hyperparameter optimization was performed. To enable explicit training with a batch size higher than one, training examples were grouped into batches by event type. To increase the stochasticity, the batches comprising same-type events in each epoch were randomly shuffled before being passed through the model. Table 3 provides the hyperparameter values for both implicit and explicit sequence labeling models.

Figure 3 Argument extraction sequence labeling on a DocEE example framed as a token-level classification problem.

Table 3 Hyperparameters for implicit and explicit sequence labeling models.

Hyperparameter	Implicit sequence labeling	Explicit sequence labeling	
Epochs	10	10	
Batch size	8	8	
Learning rate PLM	1e − 5	5e − 5	
Learning rate scheduler	Multiplicative, factor 0.99	Multiplicative, factor 0.99	
Learning rate embedding	Grid search over 5e−5,1e−5	None	
Embedding dimension	Grid search over 10,50,100	None	
Padding	Longest truncated example in batch	Longest truncated example in batch	

For the QA-based AE models, a random sample consisting of 10% of the training set was used as a validation set for early stopping during model training. The batch size was fixed to 64 questions for all models. Because the maximum number of tokens in a question is 18, we reserved 480 tokens for text and 32 tokens for questions. As 99% of arguments contain no more than 25 tokens, we limited the maximum number of tokens per answer to 30. The models were trained using the Adam optimizer (Kingma & Ba, 2014).

Results

The AE pipeline was evaluated in two ways: using ground-truth labels and predicted labels. The former constitutes an idealized setup that we include to determine the ceiling performance of the AE models. In practice, of course, the ground-truth labels are not available, and the second-stage model has to rely on the predictions of the first-stage model. Thus, AE evaluation with predicted event-type labels offers a more realistic performance estimate. The results are shown in Tables 4 and 5 for the DocEE and ASHNEE datasets, respectively.

The experiments with predicted labels were performed based on the predictions of the best-performing ED model RoBERTa (Large) (cf. Table 1). We evaluated both sequence labeling and QA-based models for AE. The “F1 (squad)” column gives the performance scores following the approach described by Rajpurkar, Jia & Liang (2018), while “F1 (exact)” shows the exact match metric.

We report the sequence labeling performance on a held-out test set from DocEE for implicit and explicit cases in terms of exact match micro F1 score and for both ground-truth and predicted labels in Table 4. We exclude sequence labeling models from ASHNEE performance since QA-based models performed considerably better in all DocEE experiments.

Table 4 shows the AE results on ground-truth event types. Models were trained and evaluated on the hard news portion of the DocEE dataset. The best-performing model is DeBERTa-v3 (Base). The sequence labeling models for AE on DocEE performed worse than the QA-based models. The implicit case worked best between the implicit and explicit sequence labeling with the embedding dimension of 100 and a learning rate of 1e − 5.

Table 4 Argument extraction results on hard news events of DocEE dataset.

The best results are in bold. For both evaluation setups, we report three numbers. The first setup evaluates argument extraction on the ground truth event types, while the second setup evaluates argument extraction on the predicted event types. The first number in F1 (squad) columns represents the micro F1 score for arguments present in the text (i.e., positive examples), while the number in the parentheses represents the overall micro F1 score –including both positive and negative examples. The F1 (exact) columns give the results of an exact match metric. The scores in F1 (squad) columns are reported only for QA-based models.

	Ground Truth (%)	Predicted (%)	
Model	F1 (squad)	F1 (exact)	F1 (squad)	F1 (exact)	
RoBERTa (implicit sequence labeling)	–	40.30	–	40.50	
RoBERTa (explicit sequence labeling)	–	38.10	–	38.00	
RoBERTa (Base, QA)	58.90 (77.90)	46.50	58.90 (77.80)	46.50	
RoBERTa (Large, QA)	58.20 (77.30)	45.50	58.20 (77.30)	45.50	
DeBERTa-v3 (Base, QA)	63.10 (79.20)	50.60	63.10 (79.00)	50.60	
DeBERTa-v3 (Large, QA)	59.50 (76.90)	45.90	59.50 (76.80)	45.90	
ALBERT-v2 (Base, QA)	58.50 (77.10)	45.80	58.50 (77.20)	45.80	
DistilRoBERTa (Base, QA)	56.40 (77.00)	44.40	56.40 (77.00)	44.40	

Table 5 Argument extraction results on hard news events of ASHNEE dataset.

The best results are in bold. For both evaluation setups, we report three numbers. The first setup evaluates argument extraction on the ground truth event types, while the second setup evaluates argument extraction on the predicted event types. The first number in F1 (squad) columns represents the micro F1 score for arguments present in the text (i.e., positive examples), while the number in the parentheses represents the overall micro F1 score –including both positive and negative examples. The F1 (exact) columns give the results of an exact match metric. The scores in F1 (squad) columns are reported only for QA-based models.

	Ground Truth (%)	Predicted (%)	
Model	F1 (squad)	F1 (exact)	F1 (squad)	F1 (exact)	
RoBERTa (Base, QA)	40.30 (70.30)	27.50	40.30 (70.60)	27.50	
RoBERTa (Large, QA)	41.70 (70.20)	28.10	41.70 (70.50)	28.10	
DeBERTa-v3 (Base, QA)	45.40 (71.30)	31.70	45.40 (71.50)	31.70	
DeBERTa-v3 (Large, QA)	42.90 (70.10)	28.30	42.90 (70.40)	28.30	
ALBERT-v2 (Base, QA)	40.10 (69.40)	26.50	40.10 (69.70)	26.50	
DistilRoBERTa (Base, QA)	37.40 (69.00)	25.30	37.40 (69.40)	25.30	

The realistic evaluation considers the pipelined model performance, i.e., the performance of the AE model on the predicted event types. For both sequence labeling and QA-based AE models, there is no significant difference between performances on ground-truth and predicted event type labels. The takeaway is that both AE models can be used on predicted event types without a noticeable drop in performance (within 1% F1). This relatively small performance drop can be explained by the high performance of ED models, which makes error propagation less likely. Also, event types have some arguments in common (those corresponding to the journalistic 5W1H questions), which perhaps contributes to a better generalization ability of the models.

Lastly, we evaluate the QA-based AE models on the ASHNEE dataset. Table 5 shows the performance of these models fine-tuned on the DocEE dataset. We observe a significant performance drop between the results of the models on DocEE and ASHNEE datasets, supporting our claim that evaluating CDMEE models on idealized datasets, such as DocEE, results in overly optimistic performance estimates. The performance of the best-performing model, namely DeBERTa (Base), drops by 17.70% in terms of the F1 (squad) metric, indicating the need for additional fine-tuning.

Accuracy-speed analysis

A real-world news event monitoring system may have to process hundreds or even thousands of documents per minute. To attain such a high document processing rate, the EE model’s inference speed (the number of predictions it can make per unit of time) must be maximized. Modern DL models are trained on GPU hardware, and the same hardware is typically also used for running the models once they have been trained. While PLMs deployed on GPUs have a much higher inference speed than models deployed on a CPU, CDMEE models differ in their architectures and the number of parameters, which results not only in differences in accuracy but also in inference speed. The inference speed also depends on the batch size (the number of documents the model processes in a single step). What this means in practice is that there exists a trade-off between CDMEE models’ accuracy and inference speed: typically, simpler models (those with fewer parameters) will have a higher inference speed but lower accuracy, and vice versa. Another consideration is that although DL models are optimized for GPU hardware, using a CPU-based infrastructure is often far more affordable, especially if one is already in place.

To determine which approach is optimal when deploying CDMEE models in production, we analyzed the trade-off between accuracy and inference speed of ED and AE models on the DocEE dataset. For the ED subtask, we define the inference speed as the average number of articles processed within one second, while for the AE subtask we define it as the average number of arguments processed within one second. We analyzed the inference speed with respect to the choice of the CDMEE model, batch size, and deployment device (CPU or GPU). All models were trained using a GPU, as that is a one-time cost compared to the inference, which is a continuous cost. To estimate the mean inference speed for ED and AE for each batch size, we ran each inference experiment 10 times and report the average throughput. Since QA-based AE models work considerably better than their sequence counterparts (cf. Table 4), we only evaluated the accuracy–speed trade-off for QA-based models.

Figure 4A shows the accuracy–speed Pareto plot for the ED models. The first noticeable effect is the stark difference in inference speed between CPU and GPU models. CPU models process at most eight articles per second, while some GPU models can process over 2,000 articles per second—an increase by three orders of magnitude. The second noticeable effect is the dependence of inference speed on batch size. We see that batch size does not influence inference speed when models are run on a CPU, but it does significantly affect the inference speed when the models are run on a GPU.

Figure 4 Comparison between F1 score and inference speed for multiple models, batch sizes, and devices (CPU, GPU) for EE subtasks.

Each model is represented with a different symbol. Batch sizes are indicated by the size of a symbol, with a bigger symbol indicating a bigger batch size. We run tests for batch sizes of 1, 16, 32, and 64 examples. The line segments indicate the Pareto front.

We conclude that when it comes to ED models, one can sacrifice a few performance points for a large gain in inference speed that comes with a smaller model, as evident by the models in the upper right corner of Fig. 4A. In particular, instead of using the best performing model, RoBERTa (Large), which can process 1,741 articles/second, one can move along the Pareto front and use the DistilRoBERTa (Base) model, which can process 2,261 articles/second. Considering that the models achieve F1 scores of 94.70% and 94.20%, respectively, we lose a mere 0.50% in classification accuracy but gain 30% in inference speed.

Figure 4B shows the accuracy–speed Pareto plot for the QA-based AE models. The first noticeable effect is again the difference in inference speed between CPU and GPU model variants. While running QA-based AE models on a CPU allows for extracting one argument per second, running the same model(s) on a GPU enables extracting up to 70 arguments per second. The second noticeable effect is that the batch size does not affect inference speed as much as for the ED subtask. Specifically, there is no difference among batch sizes of 16, 32, and 64 arguments per batch.

The best-performing model, DeBERTa (Base), achieves F1 of 61.00% and can extract five arguments per second. One can now move along the Pareto front and trade off inference speed for a decrease in extraction accuracy. Specifically, we can choose between two models, namely RoBERTa (Base) and DistilRoBERTa (Base). The RoBERTa (Base) model achieves an F1 score of 56.00% and can extract 46 arguments per second. Choosing the RoBERTa (Base) model instead of DeBERTa (Base) leads to a loss of 8% in extraction accuracy but a gain of 920% in inference speed. On the other hand, choosing the DistilRoBERTa (Base) instead of DeBERTa (Base) leads to a loss of 11.50% in extraction accuracy but at a remarkable gain of 1360% in inference speed.

If the lack of hardware resources is the main bottleneck for deploying a news event monitoring system, a potential remedy is to extract only the arguments that correspond to the 5W1H questions. Limiting argument extraction to six instead of an average of nine, and in the worst case, 20 arguments, can allow us to run argument extraction on a CPU and still extract the most relevant information about the main event.

Discussion

Implications

Our first research question (RQ1) considered the performance of state-of-the-art NLP models for CDMEE from hard news articles, looking separately at ED and AE subtasks. For the ED subtask, since both linear SVM and distilled PLMs achieve almost identical performance as larger PLMs (cf. Tables 1 and 2), we conclude that a linear SVM and distilled PLMs can be an alternative to larger PLMs. Experiments for AE (cf. Tables 4 and 5) showed that the extractive QA framework is better suited for this task than the sequence labeling framework. Moreover, we showed (cf. Subsection Argument extraction) that, in the pipeline setting, the AE model performance does not suffer from errors propagated from the best-performing ED model. Our findings indicate that the pipeline setting is suitable for the production environment. Specifically, by choosing the best-performing models from ED and AE tasks, we can achieve a high F1 score of 94.70% for ED and a respectable F1 score of 63.10% for AE. Overall, in the pipeline setting, the performance bottleneck is the AE model, but one can increase its precision at the cost of decreasing recall by discarding the low-confidence predictions.6

The second research question (RQ2) considered the problem of CDMEE performance evaluation in a realistic setup, where a model is applied to hard news articles and other articles, including soft news and event-less articles. First, we showed that ED models generalize well to noisy, automatically scraped articles, while QA-based AE models suffered a significant drop in performance, suggesting there is room for improvement and additional fine-tuning when dealing with noisy data. Furthermore, we showed that with only a slight degradation in performance, soft news articles can be considered event-less, i.e., treated as if they belong to the class other. Future labeling efforts can, therefore, safely put soft news articles in the same category as event-less articles, thus saving on labeling costs or investing in the labeling of more data.

Finally, the third research question (RQ3) considered the typical trade-off between accuracy and inference speed when using different models. We showed that the EE pipeline can be employed on both CPU and GPU hardware. By deploying the EE pipeline on a GPU, we can trade marginal performance improvement on ED for a large gain in inference speed. On the other hand, deploying the EE pipeline on the CPU is an option if one (1) employs distilled PLMs, (2) restricts the argument extraction to 5W1H questions, and (3) is not concerned with the throughput of the EE pipeline. Deploying the EE pipeline on a CPU will considerably reduce the pipeline’s throughput and render the extracted event arguments less detailed, but it may also reduce the cost and expenditure associated with acquiring GPUs.

Taken together, our results indicate that the optimal CDMEE pipeline of a hard news event monitoring system should comprise three parts: (1) a news article classifier trained to distinguish between hard news articles and all other documents, (2) an ED model based on a PLM, specifically RoBERTa, that classifies hard news articles into event types, (3) an AE model based on extractive QA and a PLM, specifically DeBERTa, that extracts arguments from the hard news article that depend on the detected event type by the ED model. We depict the proposed system architecture in Fig. 1. Our results also indicate that one can deploy ED and AE models restricted to 5W1H arguments on a CPU but at a reduced pipeline throughput compared to a GPU-based pipeline.

Limitations

Our study has two main limitations. First, we restricted the study to the English language and a monolingual setup. However, cross-lingual models have been shown to improve the performance and generalization of classification and machine translation tasks (Conneau & Lample, 2019). A cross-lingual setup has also proven successful for EE (Mutuvi et al., 2020; Mutuvi et al., 2021), improving the performance and also enabling the use of EE in low-resource settings (Hsi et al., 2016). The second limitation is the lack of additional optimizations for model inference. Those include (1) task-specific distillation, (2) downcasting of model weights, (3) quantization of model weights, and (4) weight pruning. All of the above could significantly improve the inference speed of CDMEE models on both CPU and GPU architectures, albeit further research would be needed to investigate whether the performance of distilled PLMs would be affected by such optimizations. We also acknowledge that our study did not consider the non-technical variables that factor into deployment costs, such as the cost of deploying and running CPU and GPU infrastructures and adopting GPU-trained models for inference on a CPU. Lastly, we note that our findings are specific to the PLMs we tested.

While our study identifies the optimal CDMEE pipeline, it does not address other practically relevant aspects of implementing such a pipeline. Specifically, any production-level implementation of the models we considered must also account for three key factors: reproducibility, replication, and deployment. Reproduction ensures that research results can be consistently repeated, confirming the model’s expected performance in real-world conditions. Replication addresses the challenge of models trained in controlled environments potentially behaving differently in similar but not identical settings, necessitating precautions to maintain consistent performance. Deployment involves transitioning models from lab conditions to production environments, where they must handle significantly increased data and load, ensuring scalability and reliability.

Conclusion

News event monitoring systems can help us make sense of urgent and critical events that affect our lives. Such systems rely on NLP to perform automatic event extraction from hard news articles. The task is best framed as CDMEE and decomposed into ED and AE. Recent advances in deep learning for NLP, particularly the development of PLMs, have made the CDMEE task more feasible. While state-of-the-art EE models are available and can, in principle, be used for CDMEE, there are practical issues related to model selection, robustness, and scaling that hinder their use in real-world applications. Our study explored these issues by systematically evaluating state-of-the-art CDMEE models for hard news articles.

Our results showed that distilled event extraction models based on PLMs are effective for both ED and AE. The two-stage pipeline implementing ED and AE as separate event extraction subtasks is robust to noisy data and error propagation, making it suitable for production. We also showed that for data-intensive applications, the models for both tasks must be run on a GPU, while applications with lower throughput requirements can be run on a CPU and switched to a GPU if needed. We believe our work will help researchers and industry professionals design more efficient, robust, and scalable news event monitoring applications based on the current state of the art.

There are several interesting directions for future work. While most work on event extraction has been done for the English language, news stories are published in thousands of languages, many of which are low-resource languages. This gives rise to the challenge of developing effective news event monitoring systems that work across language barriers. Recent advances in cross-lingual and multilingual deep learning models might be used to this end. Moreover, in the context of improving the scalability of event extraction models, future research could look into additional optimization techniques for inference performance. A more comprehensive deployment costs analysis could also include other non-technical variables, such as the costs of running CPU and GPU infrastructures.

Supplemental Information

Supplemental Information 1 Code

Supplemental Information 2 ASHNEE dataset

Additional Information and Declarations

Competing Interests

Author Contributions

Data Availability

1 Not to be confused with the DocEE dataset of Tong et al. (2022).

2 The DocEE does not include hard/soft event labels but merely reports the total number of each category. We analyzed and declared a total of 27 DocEE event types as hard.

4 We did not find any articles of Tsunamis event type.

5 Not all character-level spans could be aligned with the token-level positions. Overall, 2.96% spans could not be aligned, which is a negligible percentage.

6 In practice, the decrease in recall could be mitigated by leveraging the redundancy in event reporting (urgent and critical events are typically reported across many sources).

The authors declare there are no competing interests.

David Dukić conceived and designed the experiments, performed the experiments, analyzed the data, performed the computation work, prepared figures and/or tables, authored or reviewed drafts of the article, and approved the final draft.

Filip Karlo Došilović conceived and designed the experiments, performed the experiments, analyzed the data, performed the computation work, prepared figures and/or tables, authored or reviewed drafts of the article, and approved the final draft.

Domagoj Pluščec conceived and designed the experiments, performed the experiments, analyzed the data, performed the computation work, prepared figures and/or tables, authored or reviewed drafts of the article, and approved the final draft.

Jan Šnajder conceived and designed the experiments, analyzed the data, prepared figures and/or tables, authored or reviewed drafts of the article, and approved the final draft.

The following information was supplied regarding data availability:

The code and dataset are available in the Supplemental Files.

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
