# Peer review of "Closed-domain event extraction for hard news event monitoring: a systematic study"

_PeerJ Computer Science, doi:10.7717/peerj-cs.2355_

## Round 0.1 · original submission · Major Revisions

We have completed the evaluation of your manuscript. The reviewers recommend reconsideration of your manuscript following major revision. I invite you to resubmit your manuscript after addressing all the comments made by the reviewers. In particular:

1. Review the abbreviation, citation and formatting issues.
2. Provide a figure describing the general system architecture.
3. Provide the suggested tables with results.
4. Address the questions regarding soft news results formulated by reviewer 4.
5. Address the question regarding GPT-based models formulated by reviewer 4.

I hope you can complete the recommendation changes in a revision of your article.

Best,
Ana Maguitman

Reviewer 1 ·

Basic reporting

The article is cleared and well structured. Easy to understand, show flow of discussion very well. However, some references cited in text not well written (this disturb the review process), such as:
1. in line 27, 28,30,34, 41,42,45, 46.
2. Please check the manuscript for more comment in this.

Experimental design

Experimental design, cleared explanation. Explain the aim of experiment, how the experiment is conducted, tools used and dataset used very well.

Validity of the findings

Very nice and shows clearly the finding, that can be refer by others in future.

Additional comments

1. well structure article
2. the author, are required to check or to determine wither the citation reference in text is part of the sentences or only reference used for the fact.
3. suggestion, the arrangement of tables/ figures should be after the explain, so that the reader will aware of the existing those tables/ figures

Annotated reviews are not available for download in order to protect the identity of reviewers who chose to remain anonymous.
Cite this review as

·

Basic reporting

The basic writeup is good enough to understanding except few of paragraphs. The proposed methodology section name in article as "DOCUMENT-LEVEL MAIN EVENT EXTRACTION" is confusing the readers that what are the researchers going to do?

There must be hierarchal and consistence structure of headings.
The results present need to be improve especially content present in tables.
The selections of words to convey message should be academia, industrial and technical. But in this article generally technical terms are being used on abstract level. It should be understandable for all kind of researcher as well.

Experimental design

The experimental design is missing visual representation of complete developed system.

Validity of the findings

No comments yet about it.
Let fix the above recommendations that will help to understand the technical details and then evaluate the validity of research work.

Reviewer 3 ·

Basic reporting

The article "Closed-domain event extraction for hard news event monitoring: A systematic study " relates to event detection and argument extraction task. The paper has the relevant details but can be improved on the following areas:
1. Natural Language Processing should have N,L,P in capital everywhere it is used. (Line 13,43)
2. The term ‘we’ is used repeatedly which can be avoided.
3. Define hard in hard news (line 12)
4. Line 58 : wide adoption of transformers..
5. Referencing style is incorrect (Author et.al., year)
6. Line no: 96,97 unclear.
7. Too many references within a single line itself (Line no 126 onwards ).
8. Repeated use of ‘we’ in section that starts with line no 107 onwards.
9. Paragraph starting with line no 159 onwards PTPCG, DEPPN, DOC2EDAG models full expansion to be used along with abbreviation.
10. Why is the DOCEE term used for toolkit and dataset simultaneously. Please read and make it clear , it is rarely seen in such a way having name conflict.
11. footnote g and h point to the same repository. Either one can be used (Page 6).
12. Unnecessary usages of ‘we’ , ‘our’ is frequent throughout the text. Use third person.
13. Line 310 tf-idf can be in uppercase, provide its expansion also.
14. In line no 382, dataset split is mentioned as 4:1. Usually 8:2 or 6:4 is used, justify the usage of 4:1.
15. The best results in table can be highlighted in bold, rather than underline.
16. In Figure 3 a and b, the points are joined into a line. Justify why it can be drawn as a line.
Many

Experimental design

The research questions are defined.
A system architecture depicting the entire workflow is missing.
The experimental settings could be more clear if written as a table, rather than including all details in text.

Validity of the findings

The implications addressing each research question is bit confusing, Please write more clearly citing results through which is it inferred.

Additional comments

The document contain too many abbreviations which hinders its readability. The use of terms 'we' and 'our' is too frequent and the language used can be improved.

Reference format is incorrect.
Please double the paper format before resubmission.

Cite this review as

·

Basic reporting

The article is a complete practice-oriented study that is of interest to practitioners involved in the development of news monitoring systems. The research questions address the key problems of the practical use of event extraction models for monitoring news in real conditions - with noisy data and with limitations in computing performance. However, a clearer positioning of work in the problem area is necessary. In this regard some questions and recommendations could be addressed to authors:
1) The authors position the work as focused on hard news monitoring. However, the concept of hard/soft news is not entirely formal; the attribution of a particular message to hard news is largely subjective (this is evident even within this article, when the authors differently identify types of events defined in the DocEE dataset with hard/soft news in comparison with dataset developers (lines 240-241)). In this regard, the question arises: what is the technical specificity of the event extraction as applied to hard news in comparison with soft news? Perhaps it would be fairer to talk about news that is “relevant" (in context of the particular monitoring task) rather than the broader and difficult to formalize category of “hard news.”
2) The argumentation given in the Introduction about the relevance of the news monitoring task in an applied sense seems somewhat unnecessary. Because, if we assume that these arguments are addressed to practitioners involved in monitoring the current socio-political agenda or other areas, then the article is too technical. And for developers of news monitoring systems, the relevance of this task is obvious a priori.
3) The authors refused to test GPT-based models, citing the fact that they do not work well in zero-shot learning mode (line 307). However, it is unclear why in case of closed domain with explicitly defined and labeled event classes we are dealing with zero-shot learning?

Experimental design

The presented results are sufficiently supported by experimental studies, on the basis of which the authors draw reasonable conclusions. However, if the authors insist on positioning the study with a focus specifically on hard news, then it is necessary to compare the results obtained with similar results when applied to soft news.

Validity of the findings

In general, the conclusions drawn in the work are sufficiently substantiated and supported by experimental data, however, it is necessary to more clearly indicate the degree of generality of the conclusions. The authors consider 3 classes of models for event extraction - shallow models, PLM and GPT-based models. At the same time, GPT-based models are excluded from consideration due to low performance in zero-shot learning mode, and from the remaining two categories, the effectiveness of several examples of models are studied in more detail. It is unclear whether the initial list of considered EE-models is exhaustive and on what basis the models subjected to experimental research in the work were selected from this list. The work examines the effectiveness of several particular instances of EE-models of considered classes, but the conclusions are extended to the entire class. It is necessary to justify the degree of validity of such a generalization. What determines the effectiveness of a particular model - its architecture, the size or the nature of the data used for its training?
As follows from the “DocEE dataset” section (lines 240-241), the identification of news as hard or soft is completely determined by the type of event reflected in it. At the same time, soft news is regarded in this work as a noise. Taking this into account, from the experimental results we can only conclude that the studied models are capable of differentiating types of events, but not differentiating hard/soft news. Could you provide some explanations on this?

Additional comments

Some typos need to be corrected:
- On line 265, apparently, there is a typo - “USA, the US, and Australia”: does it mean UK?
- Line 603 – closing parenthesis missing.

---

## Round 0.2 · accepted · Accept

Thank you for your contribution to PeerJ Computer Science and for systematically addressing all the reviewers' suggestions. We are satisfied with the revised version of your manuscript and it is now ready to be accepted. Congratulations!

·

Basic reporting

The article is a complete practice-oriented study that is of interest to practitioners involved in the development of news monitoring systems. The research questions address the key problems of the practical use of event extraction models for monitoring news in real conditions - with noisy data and with limitations in computing performance. The authors position the results of the study for use in monitoring hard news. At the same time, they note the complexity of the formal division of hard and soft news and rely on the definition of hard news as "something that needs to be reported immediately." In this regard, a more accurate positioning of the area of application of the results of the work would be broader class of real-time news monitoring systems. Nevertheless, this remark does not reduce the significance of the work and the validity of the results obtained in it.

Experimental design

The presented results are sufficiently supported by experimental studies, on the basis of which the authors draw reasonable conclusions.

Validity of the findings

The conclusions drawn in the work are sufficiently substantiated and supported by experimental data. One of the significant results of the work is a labeled dataset suitable for use in further research on the relevant issues. In their study, the authors consider a limited, but the most relevant range of models to date, which makes the results relevant for modern researchers and practitioners in the field of intelligent news monitoring systems.

Additional comments

The article is written in clear and unambiguous, professional English.